# S'Wipe: user-friendly stool collection for high-throughput gut metabolomics and multi-omics

Dana Moradi,[1] Ali Lotfi,[1] Alexey V. Melnik,[1,2,3] Aleksandr Smirnov,[2] Konstantin Pobozhev,[2] Hannah Monahan,[2] Evguenia Kopylova,[2,4] Yanjiao Zhou,[5] Alexander A. Aksenov[1,2,3]

**ABSTRACT**  The microbiome is increasingly recognized as a key factor in health. Intestinal microbiota modulates gut homeostasis via a range of diverse metabolites. In particular, molecules such as short-chain fatty acids (SCFAs), the microbial fermentation products of dietary fiber, have been established to be reflective of microbiome and/or dietary shifts, and SCFAs alterations have been linked to multiple gastrointestinal disorders, from cancer to colitis. Despite their potential as biomarkers, technical challenges in stool collection have limited clinical translation. Here, we present Stool Wipe (S'Wipe), an ultra-low-cost fecal collection method using lint-free, mass spectrometry (MS)-compatible cellulose wipes as toilet paper. Specimens are preserved in ethanol without refrigeration and can be shipped via regular mail. Mass spectrometry analysis demonstrated that S'Wipe captures both volatile and non-volatile metabolites with reproducibility and stability validated for diagnostically relevant molecules. We show that S'Wipe performs equivalently to direct stool collection, enabling interchangeable use and comparison with existing studies. This methodology is ideally suited for large-scale population studies, longitudinal tracking, and personalized medicine applications.

**IMPORTANCE**  Gut microbiome and intestinal metabolome present invaluable diagnostic and therapeutic targets. However, conventional stool testing has several barriers, limiting bioassessment from populations. Routine, high-temporal-resolution monitoring of stool metabolome, including extensively validated and broadly informative biomarkers such as short chain fatty acids (SCFAs), is not implemented due to relatively high cost and inconvenience of sampling, possible need for clinical setting for sample collection, difficulty in collecting samples reproducibly—especially due to potential for user errors—requirement for freezer storage and maintenance of the cold chain during shipment. We present a sampling strategy specifically designed to overcome these obstacles. We demonstrate how this method can enable capturing accurate molecular snapshots at massive scales, at ultra-low cost. The approach collapses complex medical-grade collection into easy self-administration. Individuals can thereby self-monitor therapeutic responses through routine metabolome tracking, including the volatilome, otherwise hindered by infrastructure restrictions. Ultimately, this sampling approach is intended to enable participatory wellness transformation through practical high-frequency self-sampling.

**KEYWORDS**  fecal metabolome, short-chain fatty acids, gastrointestinal diagnostics, sample collection, gas chromatography-mass spectrometry, liquid chromatography-mass spectrometry, disease biomarkers, personalized medicine, metabotyping, health monitoring, patient self-testing, low cost, economical, personalized nutrition

Address correspondence to Alexander A. Aksenov, aaksenov@uconn.edu.

Dana Moradi and Ali Lotfi contributed equally to this article. The first author was primarily responsible for the manuscript revision process.

A.A.A. and A.V.M. are founders of Arome Science Inc. and BileOmix Inc. E.K. is a founder and director of Clarity Genomics. The authors declare that they have no other competing interests.

The gut microbiome plays a central role in human health in profound and wide-ranging ways through nutrient metabolism, immune modulation, and metabolic regulation, including production of bioactive metabolites (1–3). From numerous investigations of gut-derived molecules, short-chain fatty acids (SCFAs) have emerged as particularly important biomarkers of intestinal health (4). These bacterial metabolites have been extensively validated as key gut health indicators, with studies demonstrating their influence on processes spanning digestion, metabolism, immunity, and even mental health (5–7). SCFAs can be measured across the gut, with fecal samples being the most accessible (8–11). SCFAs are predominantly produced through fermentation of dietary fibers, subsequently permeate throughout the body, and affect a diverse range of tissues and physiological processes (11–13). The clinical potential of these microbial metabolites as measurable biomarkers continues to be unraveled across investigations—from IBD, all the way to the inflammatory cascade and metabolic syndrome underpinning contemporaneous epidemics of obesity and cardiovascular disease afflicting aging populations (14–18). SCFA deficiencies or disbalances have been associated with low-fiber intake diets, altered gut permeability, and inflammation, thereby promoting insulin resistance, elevating risk factors, and increasing the probability of cardiovascular events (19–22). SCFA quantification and profiling offer clinical opportunities for non-invasive monitoring, not only for disease diagnostics but also for potentially early risk assessment and exposure-response monitoring, as well as guidance for nutritional or therapeutic gut microbiome modulation. Still, SCFAs represent just one class of informative gut metabolites. Other molecular markers detectable in stool include bile acids, which reflect lipid metabolism and gut barrier function; p-cresol and indole, which indicate protein metabolism and are linked to various health conditions, such as autism spectrum disorder (ASD) (23); and a range of vitamins, amino acids, and lipids that provide insights into both dietary habits and microbial activities. Together, these diverse molecular families create a comprehensive window into intestinal health status.

Molecules related to the gut microbiome show clear associations with various health conditions; however, translating these findings into clinical biomarkers remains challenging due to high biological and technical variability. Clinical implementation requires robust reference ranges and validated quantitative associations, rather than general correlations. It is essential to distinguish consistent molecular signatures from background variability. However, current sampling methods present significant practical barriers to data collection. Fecal samples rapidly degrade under ambient conditions due to chemical degradation and microbial growth. Additionally, the reliance on specialized collection and handling infrastructure, combined with the cost and complexity of collection protocols, creates substantial logistical challenges. Non-standardized sampling protocols across studies further compound these difficulties. Consequently, currently, there is no broadly accessible, reliable personalized monitoring of gut molecular biomarkers. Finally, and perhaps most importantly, fecal collection is objectionable to many people and thus deters many individuals from participating in studies requiring fecal testing, particularly if frequent longitudinal sampling is required.

Current stool sampling collection methodologies for fecal sampling for microbiome and metabolomics analysis used in clinical practice are predominantly traditional stool collection kits consisting of stool hats, swabs, or small containers (24–30). Typically, stool needs to be collected in a screen and then manually transferred into a storage container (e.g., Cologuard test). For some methods, samples must be immediately frozen and then transported frozen to a lab to help preserve their integrity. This can lead to high variability due to differences in storage and transportation temperature, transit time, and exact collection methods between patients. An example of the currently used method is the fecal occult blood test (FOBT) cards (31–35). In FOBT sampling, fecal smears are collected at home on specialized sample cards designed to detect blood. Those cards help stabilize some molecules for transport at room temperature (RT). However, drying alters the molecular composition. Also, the small sample size and difficulties in using and handling cards limit broader profiling. Specialized solutions such

as the Cologuard can be used for non-invasive tests that analyze stool samples patients collect on a commercial kit with a preservative buffer. This test uses genetic sequencing to detect colorectal precancers but is not optimized for metabolomics (34–36). For the majority of stool-based tests (screening for blood in the stool, stool culturing for bacterial or viral infections diagnostics, ova and parasite exam, fat malabsorption assessment, etc.), stool is collected in research labs or hospitals, where clinicians may collect fecal samples directly, or during procedures like lavage or endoscopic biopsies. Obviously, such invasive techniques limit practical scalability (34).

Overall, sampling approaches that are scalable, patient- and consumer-friendly, and preserve integrity during non-refrigerated transportation have remained an unmet need. Traditional collection protocols that rely on immediate freezing are currently the state of the art (24, 26) but are logistically impractical for population-level studies. To bridge this gap, we developed the Stool Wipe (S'Wipe), a user-friendly, ultra-low-cost fecal collection method. A cellulose collection paper is used as lavatory towels, which makes sampling identical to a regular bathroom routine. Using gas chromatography-mass spectrometry (GC-MS) and liquid chromatography-mass spectrometry (LC-MS), we demonstrate that S'Wipe highly reproducibly captures native SCFAs and other diverse diet-/host-associated metabolites, without cold storage. The DNA is also captured on the cellulose and can be used for complementary sequencing. The S'Wipe method is designed to enable democratized, scalable gut molecular monitoring across global populations.

## RESULTS

We have conducted several parallel studies to test and validate different aspects of the S'Wipe methodology (Fig. 1).

### Pilot study: initial S'Wipe feasibility

To establish the feasibility of S'Wipe for metabolomics analysis (Fig. 1, Study 1), we first optimized the extraction protocol and confirmed the capture of target metabolites. As S'Wipe allows for measuring the stool weight, the abundances of metabolites could be normalized (Fig. 2A; see Materials and Methods, "Stability study"). Also, with the use of external calibration, absolute quantities for various molecules could be calculated and used as biomarkers to gain lifestyle or other information (see Fig. S2A and B at https://figshare.com/articles/figure/Figure_S2_/28404800) (see Materials and Methods, "Interpersonal variability study"). The ratios of metabolites are not dependent on the absolute quantitation and can also be linked to specific conditions (37–39, 40–44) (see Fig. S2D at https://figshare.com/articles/figure/Figure_S2_/28404800).

We primarily focused on three SCFAs with established diagnostic importance—acetic acid, propanoic acid, and butyric acid—and also other SCFAs and p-cresol (Fig. 2C; Table S1; see Fig. S6 at https://figshare.com/articles/figure/Figure_S6/28404884 and Fig. S7 at https://figshare.com/articles/figure/Figure_S7/28404908; Table S2). The comparison revealed that S'Wipe captured these metabolites with comparable or better sample-to-sample consistency than traditional methods (see Fig. S8 at https://figshare.com/articles/figure/Figure_S8/28404911?file=52314176). Statistical analysis showed an average relative standard deviation of 23.48% across all time points (see Materials and Methods, "Stability study"). While the measured concentrations aligned well across all three collection methods, S'Wipe demonstrated lower variability in replicate measurements (45) (Fig. 2D). One notable exception was acetic acid, which showed high variability across all methods, ostensibly due to its high volatility.

### Stability study: room temperature storage validation

To evaluate whether samples remain stable during non-refrigerated storage and shipping (Fig. 1, Study 3), we assessed SCFA stability over multiple days at room temperature. GC-MS analysis revealed notable stability of all nine SCFAs over the study period—no noticeable degradation was observed during room temperature storage (Fig. 2B; see

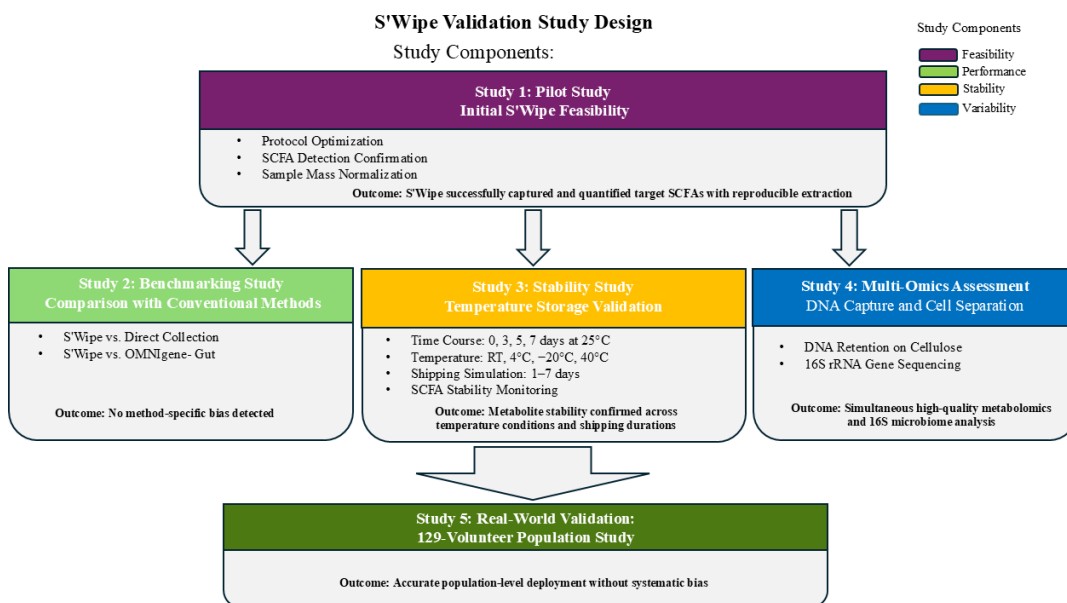

**FIG 1** The layout of studies for S'Wipe technology validation and benchmarking.

Fig. S6 at https://figshare.com/articles/figure/Figure_S6/28404884). GC-MS analysis indicated negligible changes, with no statistically significant degradation trends over the entire time interval ($P > 0.24$) of multiple days at ambient temperature (Fig. 2B). SCFA stability over 7 days at RT (Fig. 2B) aligns with Giampietro et al., who validated 95% ethanol (EtOH) for 1-week RT preservation using the Matrix Method workflow (46). Quantitative linearity of short-chain fatty acid detection spanned physiologic concentrations (47, 48) (see Fig. S3 at https://figshare.com/articles/figure/Figure_S3_/28404812). Absolute concentrations aligned with values reported from conventionally collected samples, confirming compositional integrity.

Table S3 shows the standard deviation for different SCFAs under each sample handling condition (SD < 20%). The shipping reflects a representative sum total of fluctuations in ambient conditions, particularly exposures to high temperatures and humidity. The measured abundances of SCFAs are consistent across all three sample handling scenarios, indicating that their degradation did not occur noticeably (Fig. 3B, $P > 0.05$). The coefficient of variation for all metabolites was found to be less than or equal to 20% among all samples (see Fig. S10 at https://figshare.com/articles/figure/Figure_S10/30334285), indicating similar abundances of SCFAs across storage conditions. Consequently, this indicates that the handling strategy is appropriate for the tested SCFAs within the outlined conditions.

## Benchmarking study: comparison with conventional methods

### Molecular distributions captured by S'Wipe

To comprehensively assess potential method-specific biases, we compared the broader metabolome captured by S'Wipe to that obtained through conventional stool collection, examining both volatile and non-volatile molecules with GC-MS and LC-MS, respectively. Principal coordinate analysis (PCoA) shows that there is a similarity of direct collection and S'Wipe, while OMNIgene Gut is significantly different. (Fig. 3A; Table S4, ADONIS2 pairwise comparisons, FDR < 0.05). We then employed molecular networking analysis (49, 50) to visualize and compare the captured molecular families between methods. Abundances of various SCFAs appear to be generally comparable and consistent for S'Wipe and conventional collection (Fig. 3C). The LC-MS analysis suggested that non-volatile biogenic molecules, including amino acids, bile acids, vitamins, and various microbial metabolites, were also captured similarly by both methods. A

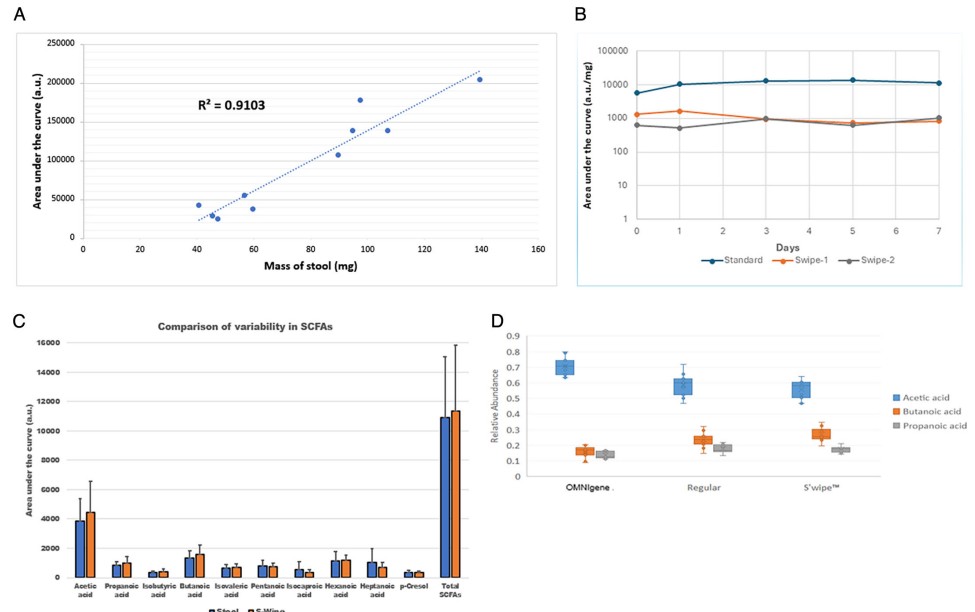

**FIG 2** Methodology benchmarking comparison. (A) Scatter plot of the total abundance of C2–C7 SCFAs vs. the measured mass of collected samples for both stool and S'Wipe. A significant positive correlation (Pearson's $R = 0.95$, $P < 0.001$) is evident. (B) Stability of SCFAs. (C) Comparison of variability in SCFAs for stool collected with the scooping method (conventional sample collection) vs. S'Wipe collection. (D) Plot showing the total concentration of SCFAs in supernatant fo three different collection methods from nine different users, normalized by the stool weight (Table S8).

notable difference between S'Wipe and direct collection was background molecules, primarily polyethylene glycols (PEGs), contributed by the collection matrix (Fig. 2C; see Fig. S4 at https://figshare.com/articles/figure/Figure_S4/28404827 and Fig. S5 at https://figshare.com/articles/figure/Figure_S5/28404872).

These findings indicate that S'Wipe collection preserves the native stool metabolome without significant alteration. After appropriate normalization and blank subtraction to account for background signals, data from S'Wipe samples should be directly comparable to direct sampling methods, as no apparent bias in metabolome composition is present.

Figure S9 (https://figshare.com/articles/figure/Figure_S9_/28404923?file=52314203) shows cell count determined by CellsBin assay (description is given in Materials and Methods, Video S1 at https://figshare.com/articles/media/Supplemental_Movies_/28462592, and Microscopy Data). It is evident that the abundance of metabolites is similar between conventional stool collection and S'Wipe, as discussed above, but the cells are generally absent in S'Wipe samples, ostensibly due to their retention on the tissue. Minimizing the presence of cell debris mitigates the risk of instrumentation contamination and circumvents issues such as clogging that may arise from excessive cellular material, enabling a simplified extraction protocol, where a single centrifugation step yields a supernatant that is ready for direct MS analysis.

## DNA and cell viability assessment

Finally, we assessed DNA capture and cell distribution to evaluate S'Wipe's multi-omics compatibility (Fig. 1, Study 4). We found that DNA was also retained on the paper material, with no detectable DNA partitioning in the supernatant. 16S rRNA gene sequencing of S'Wipe material after decanting the supernatant revealed that this collection mode efficiently captures microbial DNA, and it appears to be effectively preserved by the ethanol buffer during storage and shipping. Also, no apparent bloom of microbial species was noticeable (i.e., unexpected distribution with overrepresentation of species such as *E. coli* that are known to bloom). Moreover, S'Wipe samples

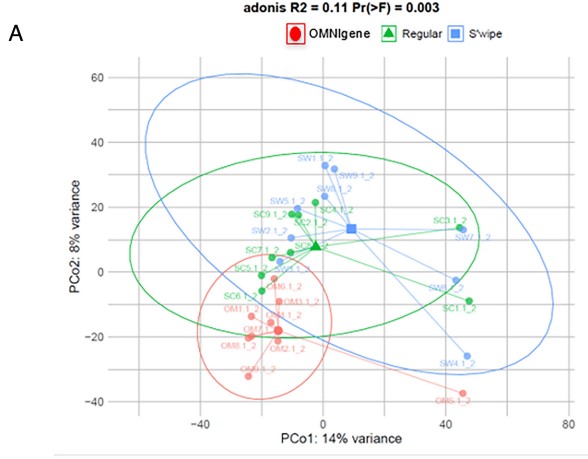

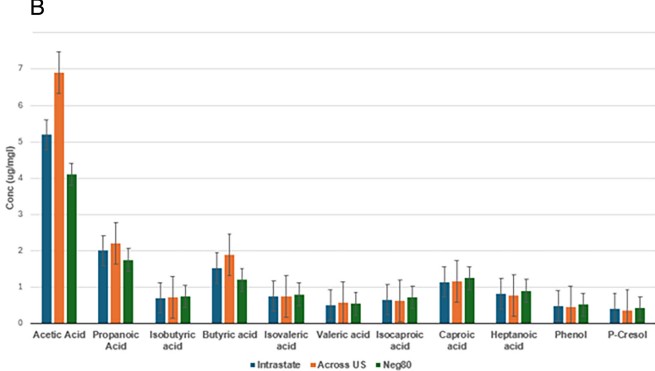

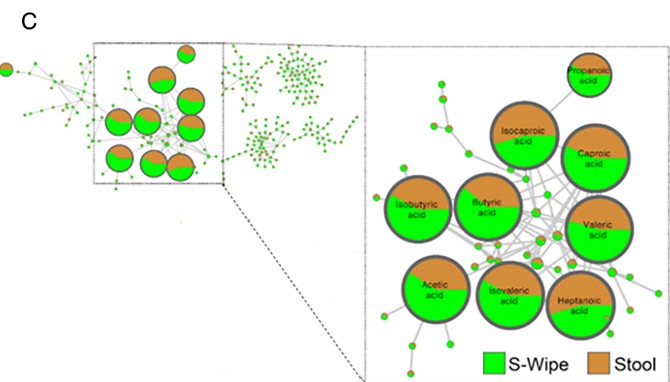

**FIG 3** Stability study. (A) Principal coordinate analysis plot of samples from the conventional stool collection and S'Wipe. Several extraction protocols have been tested, as described in Materials and Methods, to account for possible biases due to the sample preparation. (B) Bar plot showing the distribution of short- and medium-chain fatty acids as well as p-cresol and phenol across three different conditions in the shipping study. Neg80: samples stored at −80°C immediately after collection. Intrastate: samples shipped within the state (approximately 1 week passed in between shipping and receiving samples). Across the US: samples shipped from the East to the West coast of the United States (approximately 2 weeks passed in between shipping and receiving samples). The "Intrastate" and "Across US" samples were stored at 2°C–4°C upon arrival at the lab until analysis. All of the samples were analyzed by GC-MS at the same time. The analysis was conducted in triplicate. (C) A portion of the molecular network of GC-MS of the data collected using S'Wipe and regular sample collection approaches; an inset shows clusters of SCFAs. The pie chart coloring for each molecule's node corresponds to an averaged abundance of the molecule across all samples collected with the corresponding method.

consistently showed higher microbial diversity compared to DNA/RNA Shield samples across all time points (Fig. 4). As expected for developing gut microbiota, infant samples showed lower overall diversity compared to adult samples. Also, both groups exhibited the same trend—S'Wipe captured slightly greater microbial diversity than DNA/RNA Shield (51). The difference was particularly pronounced in adult samples, where S'Wipe detected several additional taxa, including members of the Erysipelotrichaceae and Christensenellaceae families. We noted the presence of some skin-associated genera, such as *Staphylococcus,* in S'Wipe samples, likely due to contact with perianal skin during collection. After excluding these skin-associated taxa, S'Wipe samples maintained higher diversity of gut-specific microorganisms compared to DNA/RNA Shield samples. The Bray-Curtis dissimilarity metrics between the two methods remained relatively stable for infant (0.17–0.21) and adult samples (0.24–0.40), suggesting consistent capture of major community structures by both sampling methods. These Bray-Curtis dissimilarity values indicate that the S'Wipe collection captures microbiome composition that is similar to the conventional DNA/RNA Shield collection (values closer to 0 indicate identical communities). For reference, values above 0.5–0.6 would typically indicate substantially different community composition, while our observed ranges of 0.17–0.21 (infants) and 0.24–0.40 (adults) suggest the alternative collection method preserves the major compositional features of the gut microbiome. These findings demonstrate that S'Wipe enables simultaneous collection of high-quality samples for both metabolomics and microbiome analyses, potentially simplifying multi-omics study designs.

## DISCUSSION

The gut represents one of nature's most complex molecular environments, shaped by dietary inputs, environmental exposures, host metabolism, and microbial chemistry. To develop reliable metabolite-based diagnostics, particularly for key diagnostic molecules like SCFAs, sampling methods that can scale to capture this complexity across large, diverse populations are needed. Current approaches are impractical for applications that demand collecting and analyzing thousands to millions of samples needed to routinely track clinical biomarkers. This provided the motivation for a simple, cost-effective stool collection method designed for population-scale metabolomic studies.

Unlike DNA sequencing advances that enabled massive initiatives like the Human Microbiome Project (53–62) and the Earth Microbiome Project (EMP) (52, 59, 60), metabolomics faces unique standardization challenges. While DNA is a consistent target molecule enabling joint co-analysis of collected data (60, 63–65), metabolites vary widely in their chemical properties, creating a bottleneck: the lack of standardized methods hinders large-scale studies, while the absence of studies comparable to EMP (53, 61) reduces pressure for methodological standardization.

A "universal" metabolomics sampling method must satisfy the following three key criteria: (i) collect native samples without introducing substantial bias, (ii) minimize distortions of molecular composition due to sample handling, storage, and preservatives, and (iii) remain compatible with various analytical platforms (GC-MS (66), LC-MS (67–71), and nuclear magnetic resonance (NMR) (63–65). Additionally, it should be cost-effective, user-friendly, and scalable, avoiding critical materials, particularly metals (53–59). These requirements differ fundamentally from DNA sequencing, where uniform protocols can be applied across all samples.

The S'Wipe collection method was designed with simplicity and the ease of use as the primary goal, building on the most basic element of the bathroom routine—toilet paper. The simplicity also makes sampling cheap (<$1 per kit) and easy to manufacture. Importantly, the approach minimizes user burden, allowing easy integration into daily routines. S'Wipe does not substantially reduce the per-sample cost of metabolomic analysis (sampling costs are relatively minor compared to downstream MS analysis), but it removes the logistical barriers that currently prevent population-level implementation. Large-scale analysis, in turn, would enable the economies of scale, which would reduce the per-sample cost. Just as commercial testing laboratories routinely process millions of

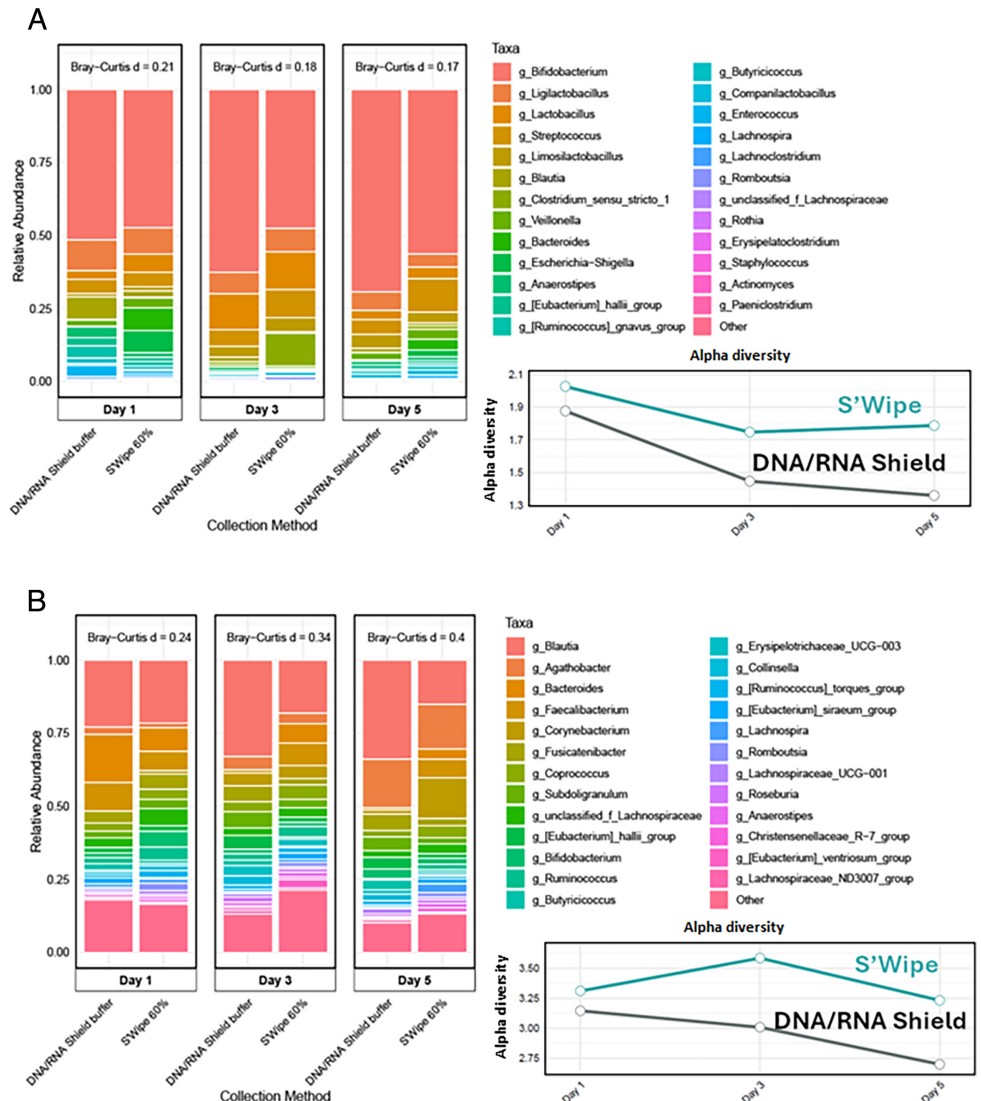

**FIG 4** 16S Top 25 genera for DNA/RNA Shield vs. S'Wipe collection. (A) Infant. (B) Adult. Each panel displays paired samples collected using DNA/RNA Shield buffer (left) and S'Wipe (right) at three independent collections from the same individuals (days 1, 3, and 5). Bray-Curtis dissimilarity values between methods are shown above each time point. Accompanying alpha diversity plots (right) show temporal changes in diversity as measured by the Shannon index (2.87–2.91 for infants and 3.5–5.2 for adults) for both collection methods (52).

clinical samples annually, metabolomics could achieve a similar scale once simplified sampling makes such widespread deployment feasible.

S'Wipe has been deployed for user gut health testing, accumulating gut metabolome statistics for a 129-volunteer cohort across multiple geographical locations (62) over several months (Fig. 5; see Fig. S12 at https://figshare.com/articles/figure/_b_Figure_S12_b_/30334303 and Table S7). This large-scale deployment serves to validate that S'Wipe captures population-level metabolite distributions, consistent with published healthy cohorts data, demonstrating the absence of systematic methodology-specific bias (72). Specifically, the observed SCFA distributions and inter-individual variability (CV = 15%–25%) align with reported values for healthy populations using conventional collection methods (reported CV ranges of 18%–30%) (72, 73). Furthermore, despite diversity in collection conditions (collections are performed by users at home), the tight variability in our cohort remains consistent with expected variations within a single controlled study, demonstrating that the method maintains standardized performance

across real-world conditions. Together with the benchmarking and stability studies, these results support S'Wipe's viability for accurate, large-scale metabolomics deployment without introducing population-level distortions.

The absence of bias also demonstrates that S'Wipe collects unaltered fecal samples, as evidenced by metabolomic profiles comparable to those obtained using traditional scooping methods (Fig. 3A; see Fig. S4 at https://figshare.com/articles/figure/Figure_S4/28404827). Correspondingly, data generated using this method should be interoperable with studies employing any conventional collection approach, provided downstream analysis protocols are standardized. This interoperability should, in principle, enable the continuation of existing studies, biobank expansion, routine sampling, and cross-laboratory data comparison. The slightly lower variability is observed for S'Wipe compared to other methods (Fig. 2D). This is expected, as volatile molecule abundances are affected by sample handling differences such as time before container sealing, ambient conditions, and sample transfer. S'Wipe samples require no additional manipulation (e.g., scooping or transfer) and are sealed almost immediately, minimizing loss of volatiles, thereby reducing variability.

Sample mass determination and normalization enable quantitative measurement of molecules of interest. The amount of collected material varies naturally across samples; sample weight is determined by subtracting the standardized weights of the collection tube and wipe from the total kit weight. The strong correlation between SCFA abundance and collected biomass (Fig. 2A) supports the utility of the mass-based normalization approach. While metabolite concentrations in whole stool samples often correlate poorly with total mass due to variations in water and fiber content, S'Wipe collection indicates a robust correlation between SCFA abundance and biomass.

The robust quantitation requires a preservation medium that ensures sample stability and is compatible with downstream metabolomics analysis. We selected 60% ethanol as a versatile preservation medium (see Fig. S11 at https://figshare.com/articles/figure/_b_Figure_S11_b_/30334291?fil and Table S6). This solvent effectively captures many diagnostically relevant metabolites, including SCFAs, while being cost-effective and of low toxicity. Ethanol suppresses microbial growth and enzymatic activity (60); our data show stable SCFA levels after several days at room temperature (Fig. 2B). While the S'Wipe protocol has been benchmarked with ethanol, it can be compatible with other solvents and be adapted for specific applications—samples can be lyophilized and reconstituted in alternative solvents to optimize the detection of particular molecular classes (this approach is only suitable for non-volatile compounds). The choice of solvent

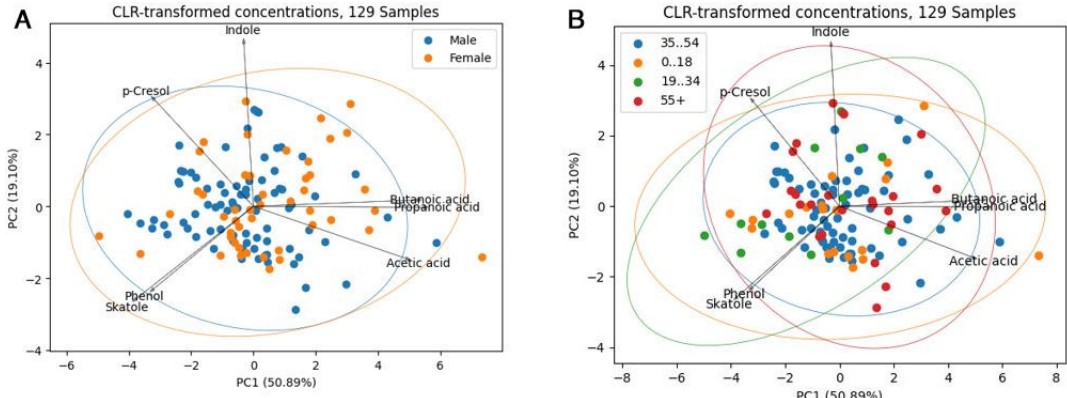

**FIG 5** Population trends. (A) Comparison of microbiota shifts between genders. (B) Comparison of microbiota shifts across age groups. PCoA plot based on CLR-transformed concentrations showing microbial community differences among four age groups: 0–18, 19–34, 35–54, and 55+. The 0–18 group shows slightly greater separation from the others, likely reflecting the developmental nature of the gut microbiome during childhood. Arrows represent metabolite loadings; beneficial short-chain fatty acids (SCFAs), such as butanoic acid, are associated with one direction of the ordination space, while potentially pathogenic metabolites, such as phenol, are oriented in the opposite direction.

can be tailored to specific analytical methods (e.g., RP vs. HILIC for LC-MS). However, when targeting molecules beyond SCFAs, their stability should be validated.

Finally, we have demonstrated that the S'Wipe method separates metabolites (in supernatant) from cells and DNA (retained on cellulose), with no cell debris and DNA detected in the supernatant (see Fig. S9 at https://figshare.com/articles/figure/Figure_S9_/28404923?file=52314203). This yields low-debris samples that simplify preparation and minimize downstream mass spectrometry (MS) issues such as column clogging, carryover, and ion suppression. Additionally, the method enables both metabolomics and microbiome analysis from a single collection event, an advantage over traditional approaches that require separate sampling protocols for each omics measurement. This would further simplify longitudinal studies requiring paired metabolomic and microbiome data and support scaling to population-level cohorts.

## Conclusion

S'Wipe is a stool collection method designed for high-frequency sampling for metabolomics analysis with the simplicity of a toilet wipe. The method demonstrates analytical sensitivity, quantitative capability, reproducibility, and molecular preservation, retaining and releasing key gut metabolites including SCFAs, bile acids, lipids, and vitamins. By simplifying stool collection, S'Wipe is designed to translate gut metabolome monitoring from an academic domain into a practical tool for personalized nutrition and disease management. Similarly, to how affordable genetic testing enabled proactive management of hereditary risks, accessible gut metabolome monitoring through routine SCFA measurement could shift gastrointestinal health management from reactive treatment to predictive intervention.

## MATERIALS AND METHODS

### S'Wipe kits

All of the conducted studies have utilized S'Wipe kits manufactured byBileomix Inc. according to ISO 9001 standard (https://bileomix.com/product-category/bileomix-s-wipe/). The kits contained lint-free paper (Kimberly-Clark Professional, 34120), the preservation solution in a 5 mL container spiked with an internal standard, a tube stand, and a sampling instruction sheet. The kits were provided to study participants free of charge.

### S'Wipe sampling protocol

The S'Wipe sampling approach consists of three key components: (i) a lint-free cellulose collection paper that serves as the sampling matrix, (ii) a collection tube containing preservation solution (60% ethanol is utilized throughout this study), and (iii) a standardized protocol for sample handling and processing. The name "S'Wipe" refers to this complete sampling system, where the collection paper is used like conventional toilet paper during normal bathroom routines and then placed in the preservation solution for storage and shipping.

With the use of S'Wipe, approximately 100 mg of stool can be typically collected. The wipe is then placed by the user into a 30 mL wide-mouth centrifuge tube with 5 mL of MS-grade 60% ethanol spiked with internal standard, sealed, and then can be both stored and/or shipped at room temperature until extraction and analysis (Fig. 6). In the proposed approach, ethanol solution is used for sample preservation. Ethanol is known to both denature enzymes, preventing enzymatic degradation (61, 62, 72–78), and kill microorganisms, preventing blooms (61, 79–82). The users are not required to modify their bathroom routine, with the exception of placing the used paper into a collection tube rather than flushing/throwing it away. Due to the simplicity of collection, sampling can be conducted anywhere, and every bowel movement could be sampled, if needed. Correspondingly, large longitudinal sampling with each subject being their own control for intervention studies can be facilitated. During shipment, the extraction

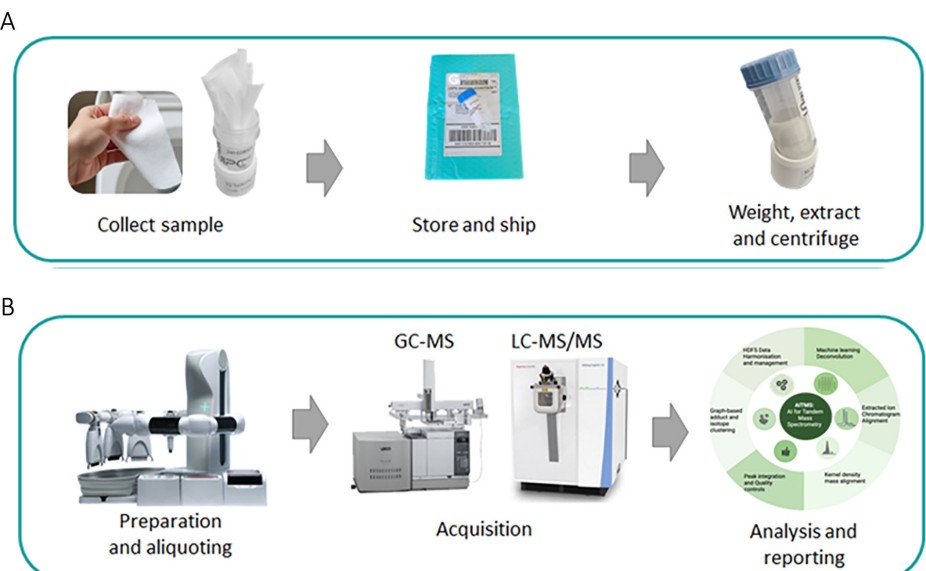

**FIG 6** S'Wipe sampling. (A) Sampling and shipping workflow using the kit (the sample collection kit is shown in Fig. S1 at https://figshare.com/articles/figure/Figure_S1_/28404761?file=52313852). (B) Schematic sample preparation and analysis workflow. The patent application is pending (U.S. Application #63/570,322).

solvent remains absorbed by the wipe and acts as a preservative. Upon arrival at the analysis lab, the samples are centrifuged to separate the supernatant from the wipe. The supernatant is directly compatible with mass spectrometry analysis, both GC-MS and LC-MS, without any additional sample preparation steps, with the exception of possible dilution. The absolute measured abundances of biomarkers of interest, as well as longitudinal trends, can then be used to inform users.

## Pilot study

A healthy volunteer provided all stool samples. The protocol used in this study was run alongside a long sonication and incubation extraction protocol. The long protocol included a 10-min sonication on ice and 6 h on a vibration table at 2°C–4°C for homogenization. Samples included S'Wipe kits with no wipe, S'Wipe kits complete, and S'Wipe kits with no stool. Samples were generated in triplicate. After the sample extraction, the samples were analyzed by GC-MS and LC-MS/MS.

Different parameters, such as the order of steps, solvent volume, centrifugation time, and setting, were tested to establish the standard protocol, which was selected as the simplest and most effective approach. The optimized protocol is as follows.

Completed S'Wipe kits kept at −80°C were thawed on ice prior to extraction. Pre-sample and post-sample kit mass was recorded. The completed kits were first placed in an iced ultrasonic bath for 10 min and then on a vibration table for 10 min at room temperature for homogenization. In total, 1,000 µL aliquot was taken and transferred to 1.5 mL microcentrifuge tubes. The samples were centrifuged for 5 min at 14,000 revolutions per minute. After centrifugation, aliquots of 100 µL were taken from each sample and transferred to vials with conical inserts and analyzed by GC-MS and/or LC-MS/MS. Samples were directly loaded onto GC-MS. For LC-MS/MS, samples were diluted 4-fold with pure cold methanol to precipitate protein. The samples were then filtered through Phenomenex Phree plates with the application of 4 psi negative pressure. In total, 200 µL of collected filtrates were dried under vacuum and reconstituted in 100 µL of C18 resuspension buffer (5% acetonitrile [Sigma, USA] in LCMS grade water) supplemented with added internal standards: sulfachloropyridazine (TCI, USA) and sulfamethazine (Sigma, USA).

The S'Wipe sampling was found to capture compounds of interest such as SCFAs, p-cresol, and other molecules that are expected to be detectable in stool, such as indole, bile acids, vitamins, amino acids, and lipids (see Fig. S2 at https://figshare.com/articles/figure/Figure_S2_/28404800 [83–91], Fig. S4 at https://figshare.com/articles/figure/Figure_S4/28404827, and Fig. S5 at https://figshare.com/articles/figure/Figure_S5/28404872 [92, 93]). While molecular networking demonstrates that S'Wipe preserves the broader stool metabolome, we specifically focused our validation on SCFAs and other established diagnostic markers to rigorously assess the method's performance for molecules of immediate clinical relevance.

## Stability study

The stability of SCFAs under room temperature storage conditions was assessed. While SCFAs are generally stable at room temperature, their preservation specifically within the S'Wipe matrix needs to be validated. A healthy volunteer donor has provided all stool samples. Two complementary experiments were conducted: first, S'Wipes were spiked with a 500 µL aliquot of Accustandard FAMQ-004 standard (New Haven, CT, USA) at a concentration of 10 mM to track the recovery of known SCFA concentrations. Second, stool samples were collected using S'Wipe to assess the stability of endogenous SCFAs. Both the standard-spiked wipes and stool-sampled wipes were stored at room temperature for periods ranging from 1 day to 1 week (Fig. 2B).

For longitudinal stability assessment, samples collected by S'Wipe kits and controls were incubated at 25°C for 0, 3, 5, and 7 days. Negative controls contained no stool, and positive controls contained no stool and Accustandard FAMQ-004 (mix of SCFAs) at 1 mM. All samples were collected in triplicate. Sampled kits were stored at −80°C until analysis, and all samples were extracted together. After the sample extraction, the samples were analyzed by GC-MS and LC-MS/MS.

To assess the stability of 10 SCFAs at different temperatures, samples were stored at room temperature, 40°C, 4°C, and −20°C over 0, 1, 2, 3, 12, 21, and 30 days. All samples were collected in triplicate, resulting in a total of 85 samples. After storage, the sample kits were placed into a −80°C freezer until analysis, and all samples were extracted at the same time (see Fig. S10 at https://figshare.com/articles/figure/Figure_S10/30334285 and Table S5).

## Shipping study

To evaluate metabolite stability during shipping, S'Wipe samples were subjected to three conditions: immediate storage at −80°C, room temperature storage for 1 week for short-distance shipping (36 miles, intrastate), and room temperature storage for long-distance shipping (2,950 miles, East to West coast, USA). Each condition was tested in triplicate via the United States Postal Service, with samples stored at 2°C–4°C upon receipt until analysis. A healthy volunteer donor has provided all stool samples. All samples were in triplicate. Upon reception in the mail, the S'Wipe kits were stored at 2°C–4°C until analysis. All samples were extracted and analyzed by GC-MS and LC-MS/MS at the same time.

## Benchmarking study

S'Wipe (using 60% ethanol) was benchmarked against conventional stool collection (direct collection of bulk stool using a scooping/transfer method into collection tubes, representing the standard laboratory approach) and OMNImet Gut (DNA Genotek, Ottawa, Canada) sampling methods. The latter was selected as a representative commercially available methodology for room-temperature sample stabilization and metabolomics sampling. Side-by-side comparisons were conducted to assess metabolite recovery and variability (Fig. 2D).

The 60% ethanol concentration buffer for S'Wipe was selected based on prior studies showing effective metabolite preservation with reduced flammability compared to 95%

ethanol (94, 95). Previous studies comparing preservation methods (95) demonstrated that 95% ethanol provides excellent metabolite preservation compared to FOBT and FIT cards, although high ethanol concentration can complicate sample handling and processing. The 60% ethanol concentration is chosen to balance preservation with practical handling considerations (particularly reduced flammability), while maintaining similar metabolite recovery to direct collection methods, as demonstrated in Fig. 2.

## Interpersonal variability study

Ten healthy volunteers were provided with S'Wipe collection kits and US mail return labels to return samples to the lab. No personal or any other information was collected throughout the study.

## GC-MS data acquisition and processing

One-microliter aliquots of the S'Wipe kit supernatant were directly injected into an Agilent 6890 GC interfaced to a mass spectrometer LECO BT MS 5973 for electron ionization GC-MS. The injector was maintained at 250°C with a split ratio of 10:1. The GC utilizes a 30 m ZB-FFAP column (0.25 mm i.d., 0.25-μm film thickness) for metabolite separation with 1.2 mL/min constant He flow. The oven temperature program initiates at 50°C, rising to 240°C at 10°C/min. No noticeable carryover was observed over the entire injection sequence for all of the studies. Also, no increased contamination that necessitates liner change has been observed, indicating that possible lint particulate traces from wipe material did not contribute to any observable analytical interference. The data were then deconvoluted with the MSHub algorithm (49). The experimental spectra were searched against the NIST 2023 library with ≥80% spectral match defining putative identifications, and the retention times falling within <0.01 min of the corresponding reference standards. Targeted analysis of SCFAs and p-cresol was performed by using Leco ChromaTof software.

Targeted analysis of SCFAs and p-cresol was performed using Agilent Mass-Hunter software. The reference concentration ranges were determined from literature cited in the Human Metabolome Database (HMDB). The targeted panel included acetic acid (HMDB0000042), propanoic acid (HMDB0000237), isobutyric acid (HMDB0001873), butyric acid (HMDB0000039), isovaleric acid (HMDB0000718), valeric acid (HMDB0000892), isocaproic acid (HMDB0000689), caproic acid (HMDB0000535), heptanoic acid (HMDB0000666), phenol (HMDB0000228), p-cresol (HMDB0001858), indole (HMDB0000738), and skatole (HMDB0000466) as annotated in the Human Metabolome Database (HMDB).

## LC-MS/MS data acquisition and processing

The samples were injected and chromatographically separated using a Vanquish UPLC (Thermo Fisher Scientific, Waltham, MA) on a 100 mm × 2.1 mm Kinetex 1.7 μM, C18, 100 Å chromatography column (Phenomenex, Torrance, CA), 40°C column temperature, 0.4 mL/min flow rate, mobile phase A 99.9% water (J.T. Baker, LC–MS grade), 0.1% formic acid (Thermo Fisher Scientific, Optima LC/MS), mobile phase B 99.9% acetonitrile (J.T. Baker, LC–MS grade), and 0.1% formic acid (Fisher Scientific, Optima LC–MS), with the following gradient: 0–1 min 5% B, 1–8 min 100% B, 8–10.9 min 100% B, 10.9–11 min 5% B, and 11–12 min 5% B.

MS analysis was performed on an Orbitrap Exploris 240 (Thermo Fisher Scientific, Waltham, MA) mass spectrometer equipped with HESI-II probe sources. The following probe settings were used for both MS for flow aspiration and ionization: a spray voltage of 3,500 V, a sheath gas ($N_2$) pressure of 35 psi, an auxiliary gas pressure ($N_2$) of 10 psi, an ion source temperature of 350°C, an S-lens RF level of 50 Hz, and an auxiliary gas heater temperature of 400°C.

Spectra were acquired in positive ion mode over a mass range of 100–1,500 $m/z$. An external calibration with Pierce LTQ Velos ESI positive ion calibration solution (Thermo

Fisher Scientific, Waltham, MA) was performed prior to data acquisition with a ppm error of less than 1. Data were recorded with a data-dependent MS/MS acquisition mode. A full scan at the MS1 level was performed with 30K resolution. MS2 scans were performed at 11,250 resolution with a max IT time of 60 ms in profile mode. MS/MS precursor selection windows were set to $m/z$ 2.0, with an $m/z$ 0.5 offset. MS/MS active exclusion parameter was set to 5.0 s.

LC-MS raw data files were converted to mzML format using msConvert (ProteoWizard). MS1 features were selected for all LC-MS data sets collected using the open-source software MZmine 3 (79) with the following parameters: mass detection noise level was 10,000 counts, chromatograms were built over a 0.01-min minimum time span, with 5,000-count minimum peak height, and 5-ppm mass tolerance; features were deisotoped and aligned with 10-ppm tolerance and 0.1-min retention time tolerance; and aligned features were filtered based on a minimum 3-peak presence in samples and based on containing at least two isotopes. Subsequent blank filtering, total ion current, and internal standard quality control were performed. To evaluate potential polyethylene glycol (PEG) contamination from the cellulose matrix, we monitored PEG signal intensities across extended injection sequences (>500 injections) during method development. PEG levels were quantified as a percentage of total ion current for each sample and compared between S'Wipe samples, conventional stool samples, and blank controls. Column performance was monitored using quality control standards injected at regular intervals throughout the sequence. No increased PEG accumulation was observed, with levels consistently remaining below 0.01% of total ion current.

## Untargeted metabolomics data processing

MetaboAnalystR (96) was used to preprocess the data by replacing zero peak abundance values with 1/5 of the smallest positive value for each feature, representing the limit of detection. Peak abundances below 1 (4.4% of data) were adjusted to 1 to prevent negative values during log transformation. This preprocessing step ensures compatibility with downstream batch correction tools (e.g., ComBat, WaveICA2) and avoids issues with negative values in the corrected matrix. Within the overall data distribution, metabolite features with such low peak abundances (<1) are generally biologically insignificant and likely stem from zeros in the raw data. Features with no variance (constant or single value across samples) were removed. Quantile normalization was performed to standardize distributions across samples to reduce technical variability, followed by centered log-ratio (CLR) transformation. PERMANOVA, implemented using the adonis2 function from the "vegan" package in R (number of permutations = 999), was performed to test whether the multivariate means of S-wipe and stool samples differed significantly, using the Euclidean distance matrix of the CLR-transformed feature table as input.

## Cell counting assay

To assess S'Wipe's compatibility with MS without extensive sample cleanup, we tested its ability to minimize cell debris in the supernatant. Fecal samples typically contain host and microbial cells that can interfere with MS analysis by causing clogging and background noise (58, 80, 81). S'Wipe was designed to retain fecal material on the cellulose matrix, allowing soluble metabolites to partition into the solvent. A comparative cell counting analysis (including both native feces and supernatant) was conducted between samples obtained using the S'Wipe approach and those collected through conventional methods using CellsBin's VEGA platform (see Results and Fig. S9 at https://figshare.com/articles/figure/Figure_S9_/28404923?file=52314203). The analysis was conducted by the device manufacturer. The normalized density of each sample was calculated by imaging 20 µL through CellsBin's microfluidic optical device. The company's machine learning algorithm detected and classified particles, estimating the relative size and normalized density of each sample.

## Stool sample collection, DNA extraction, and 16S rRNA V4 region sequencing

The potential dual metabolomics/microbiome analysis using S'Wipe was evaluated. Samples from both adult and infant subjects were provided from three time points (days 1, 3, and 5) and compared against samples collected using DNA/RNA Shield buffer, a widely used commercial solution optimized for microbiome preservation and DNA extraction (Fig. 4).

The sampled tissue wipes were transferred into 50 mL Falcon tubes pre-filled with 60% of ethanol or DNA/RNA Shield. A blank control tube was also included, containing DNA/RNA Shield but no wipe with the stool sample, to monitor potential contamination from the tissue wipe or reagents. All samples were stored at 4°C. To extract DNA from the wipe samples, the tissue wipe remained in the Falcon tube, and a 1 mL pipette tip was used to press the wipe repeatedly to extract the liquid absorbed within it. This process yielded approximately 100–300 µL of liquid per sample. The extracted liquid was then centrifuged at $15,000 \times g$ for 1 min, resulting in the formation of a visible stool pellet at the bottom of the tube. This pellet was subsequently used for DNA extraction using the Qiagen PowerSoil DNeasy Pro Kit. The V4 region of 16S rRNA was amplified and sequenced at the Illumina MiSeq platform.

## Real-world population validation

### Study design

To validate S'Wipe performance under real-world conditions and assess population-level metabolite distributions, samples were collected from 129 volunteers across multiple geographic locations using home-based collection protocols. This deployment aimed to demonstrate method robustness in uncontrolled settings and confirm that S'Wipe captures physiologically relevant metabolite patterns without systematic bias.

### Data preprocessing and normalization

Concentrations of seven short-chain fatty acids and microbial metabolites (acetic acid, butyric acid, propionic acid, p-cresol, indole, skatole, and phenol) were quantified via GC-MS analysis. Samples with missing concentration values or quantification errors were excluded, yielding a final data set of 129 samples with complete observations across all seven compounds. Demographic metadata, including sex and age, was categorized into age groups (0–18, 19–34, 35–54, and 55+ years).

To account for the compositional nature of metabolite concentration data (where values are constrained by total sample mass), concentrations were transformed using the centered log-ratio (CLR) method. CLR-transformed values were then standardized to unit variance for each compound to ensure equal weighting in multivariate analysis.

### Principal coordinate analysis

To visualize metabolite profile similarities across the cohort, Bray-Curtis dissimilarity matrices were calculated from the preprocessed metabolite concentrations. PCoA was performed on the dissimilarity matrix to reduce dimensionality. The first two principal coordinates (PCo1 and PCo2) of the total variance were extracted for visualization. Samples were colored by demographic groups and overlaid with 99.7% confidence ellipses (three SDs) to visualize group clustering patterns. Metabolite loadings were scaled by their explained variance contribution and displayed as vectors to indicate their influence on the principal coordinates.

## ACKNOWLEDGMENTS

The authors appreciate the help and materials in the form of automation support provided by TomTec Inc. (http://www.tomtec.com/). Additionally, the authors thank CellsBin Inc. (https://www.cellsbin.com/) for the analysis of samples for microscopy measurements (CellsBin, based in New Haven, CT, is a tech-bio company; its VEGA

platform is utilized in precision oncology and other fields, including aerosol detection for the IARPA's Picard Project [https://www.iarpa.gov/research-programs/picard]). Arome Science Inc. provided funding for the study.

A.V.M. and A.A.A. formulated the study. A.V.M., K.P., and H.M. prepared the samples. A.V.M. devised the sampling manifold. A.V.M., K.P., and A.L. performed the GC-MS and LC-MS analysis. E.K., A.V.M., D.M., A.L., and A.S. conducted data processing and statistical analysis. A.A.A., A.V.M., and H.M. coordinated the study. A.A.A., D.M., A.L., and A.M. wrote the manuscript. All authors read and approved the manuscript.

## AUTHOR AFFILIATIONS

[1]Department of Chemistry, University of Connecticut, Storrs, Connecticut, USA
[2]Arome Science Inc., Farmington, Connecticut, USA
[3]BileOmix Inc., Farmington, Connecticut, USA
[4]Clarity Genomics Inc., San Diego, California, USA
[5]UConn Health, Farmington, Connecticut, USA

## AUTHOR ORCIDs

Dana Moradi http://orcid.org/0009-0000-2088-485X
Ali Lotfi http://orcid.org/0000-0001-6124-3338
Alexey V. Melnik http://orcid.org/0000-0002-4645-8880
Alexander A. Aksenov http://orcid.org/0000-0002-9445-2248

## DATA AVAILABILITY

All data generated in this study are publicly available. The raw data are available on MassIVE Repository (massive.ucsd.edu) under the following data set accession numbers: MSV000094530 (GC-MS), MSV000094529 (LC-MS), and MSV000096969. Reproducibility data (GC-MS) GNPS job links are as follows: https://gnps.ucsd.edu/ProteoSAFe/status.jsp?task=eab1df9e88584bf289969bceb0ca59a3; https://gnps.ucsd.edu/ProteoSAFe/status.jsp?task=bf75f44dfab149248e66fafa6dd74e0a.

## ETHICS APPROVAL

All samples analyzed in this study were collected by voluntary participants who provided informed consent for their de-identified data to be used for research purposes and publication. As this involved retrospective analysis of de-identified data from a minimal-risk commercial activity, this work was exempt from IRB review under 45 CFR 46.104(d)(4).

## ADDITIONAL FILES

The following material is available online.

### Supplemental Material

**Microscopy Data (mSystems01459-25-s0001.xlsx).** CellsBin assay for samples 1 to 6 and 10 to 12.
**Legends (mSystems01459-25-s0002.docx).** Legends for supplemental tables.
**Table S1 (mSystems01459-25-s0003.docx).** Comparison of three collection methods.
**Table S2 (mSystems01459-25-s0004.docx).** The $P$ value SD within stool samples and SD S'Wipe for each SCFA.
**Table S3 (mSystems01459-25-s0005.docx).** Standard deviation for Neg 80, Across US, and intrastate handling conditions.
**Table S4 (mSystems01459-25-s0006.docx).** Principal coordinate analysis results for comparisons of S'Wipe, direct collection, and OMNIgene Gut.

**Table S5 (mSystems01459-25-s0007.docx).** Coefficient of variation, standard deviation LOD/LOQ, and curve-based LOD/LOQ results for stability of ten SCFAs at different temperatures and times.

**Table S6 (mSystems01459-25-s0008.docx).** Comparison of extraction reproducibility across different ethanol concentrations.

**Table S7 (mSystems01459-25-s0009.docx).** Coefficient of variation results of seven SCFAs for 129 samples to reveal reproducible performance across diverse populations.

**Table S8 (mSystems01459-25-s0010.docx).** *P* values for acetic acid, butanoic acid, and propanoic acid comparing different collection methods.

## Open Peer Review

**PEER REVIEW HISTORY (review-history.pdf).** An accounting of the reviewer comments and feedback.

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
