## [Reviewer comments · mSystems]

S'Wipe: User-Friendly Stool Collection for High-Throughput Gut Metabolomics

Dana Moradi, Ali Lotfi, Alexey Melnik, Aleksandr Smirnov, Konstantin Pobozhev, Hannah Monahan, Evguenia Kopylova, Yanjiao Zhou, and Alexander Aksenov

Corresponding Author(s): Alexander Aksenov, University of Connecticut

Review Timeline:

Submission Date:	October 21, 2025
Editorial Decision:	December 1, 2025
Revision Received:	January 22, 2026
Accepted:	January 27, 2026

Editor: Marnix Medema

Reviewer(s): The reviewers have opted to remain anonymous.

Transaction Report:

DOI: <https://doi.org/10.1128/msystems.01459-25>

Re: mSystems01459-25 (**S'Wipe: User-Friendly Stool Collection for High-Throughput Gut Metabolomics**)

Dear Dr. Alexander A Aksenov:

The manuscript has been seen by one of the previous reviewers, and, because the other was not available, by one fresh reviewer. The previous reviewer is largely satisfied, but the new reviewer raises some additional points regarding clarity of the storyline and relations between the experiments performed. My recommendation regarding these points would be to textually clarify the storyline and the rationale / relationships between several analyses to make the manuscript more compelling for the mSystems audience; I do not think large amounts of new analyses are required. Yet, please do also address the remaining technical comments. I look forward to seeing a revision, which, if the points are addressed well, I may then be able to assess without requiring another round of reviews.

Revision Guidelines

Sincerely,
Marnix Medema
Editor
mSystems

Reviewer #1 (Comments for the Author):

The authors revised the manuscript substantially again. The authors, for the most part, addressed my comments very well!

The one remark I have regarding comments raised by me, relates to lines 228-231 - where do those reference ranges come from? Were they established by the authors (how?), otherwise this claim should be supported by a reference.

Since population trends analysis (Figure 4 and corresponding text) have been added to the manuscript, I would also like to comment on that.

As it stands, I do not understand what this follow-up analysis is supposed to achieve. More people (129 volunteers) are included in the study and their results are presented in Figure 4 but it is not explained what this shows/achieves. There are no control samples included or references to literature data which would somehow prove or illustrate that the S'Wipe results more robust than with other collection devices (a controlled study with 2 collection devices), or that the users somehow prefer S'Wipe over other collection methods (a survey?).

Reviewer #3 (Comments for the Author):

Synopsis

The manuscript entitled "S'Wipe: User-Friendly Stool Collection for High-Throughput Gut Metabolomics" by Moradi et al. describes the validation of a new technique for sampling and stabilizing fecal samples. In brief, the method uses a Kim-wipe to sample stool by having the donor wipe their anus following defecation then transferring the soiled Kim-wipe to a tube containing ethanol. The authors compared this method to sampling bulk stool and using another commercially available sampling method. The authors used GC-MS and LC-MS to explore the volatile and non-volatile metabolomes of stool to compare to other sampling methods.

Reviewer comments:

- The title is misleading, the authors state this is metabolomics however they are including RNA analysis and cell counting assays. I suggest the authors either remove these sections and resubmit to another journal or change the title.
- The authors are missing recent literature in stabilizing stool samples. Additionally, the authors have not discussed their results in the context of the broader literature. They need to perform a deeper analysis of the literature, and discuss results in the context of up to date literature.
- The studies performed are disjointed and confusing. The authors need a figure in the materials and methods which summarizes the studies performed to allow the reader to follow the individual experiments. Additionally, these experiments are described separately in the methods but are not consistently labeled in the results. I suggest the authors use same or similar headings from the methods to discuss the results.
- The authors fail to discuss normalization beyond the ethanol containing internal standard. I am concerned that the mass of stool sampled is going to be highly variable between participants due to differences in their anatomy and dietary habits. This needs to be discussed further.
- Overall feeling is this manuscript contains multiple experiments and feels disjointed.
- Authors need to restructure results to match experimental headings before manuscript is suitable for publication.
- The 3rd sentence of the importance statement is a run-on sentence.
- References in text should be after punctuation.
- Principal coordinate analysis is unusual for metabolomics. Looking at how far apart samples are rather than how they are related linearly. Please justify the use of PCoA or use PCA.
- Methods lacking details on how data were preprocessed and normalized for PCoA and other statistical analysis.
- Discussing trace polyethylene glycols (PEGs) in Kimwipes is concerning for LC-MS, as PEGs are extremely difficult to remove from the column and will concentrate, especially in large cohort high throughput studies. Please discuss this further.
- Using ml and µl when should be mL and µL.
- Define conventional stool collection.
- GC-MS methods missing inlet temp and split information.
- Log transformation of metabolomics data is unusual unless the data are explicitly on a log scale. Typically autoscale is performed based on how the data are distributed.
- It is worth noting that DNA Genotek has a tube that is designed for metabolite stabilization.
- Authors are missing ethics for human sample collection.

Dear Marnix,

We are grateful to you and the reviewers for the thorough assessment of our manuscript. We have addressed all of the comments through the following key revisions:

1. We further improved study organization and added a study design schematic (new Figure 2) to provide clear logical flow between all of the individual validation/benchmarking studies.
2. We added literature comparisons demonstrating that S'Wipe captures population-level SCFA distributions consistent with published values, validating the absence of methodology-specific bias.
3. We modified the title to reflect multi-omic scope and clarified participant consent procedures.

We believe these revisions have directly addressed the concerns that Reviewers brought up. Below are our point-by-point responses to all comments.

Sincerely,

Alexander

Dear Dr. Alexander A Aksenov:

The manuscript has been seen by one of the previous reviewers, and, because the other was not available, by one fresh reviewer. The previous reviewer is largely satisfied, but the new reviewer raises some additional points regarding clarity of the storyline and relations between the experiments performed. My recommendation regarding these points would be to textually clarify the storyline and the rationale / relationships between several analyses to make the manuscript more compelling for the mSystems audience; I do not think large amounts of new analyses are required. Yet, please do also address the remaining technical comments. I look forward to seeing a revision, which, if the points are addressed well, I may then be able to assess without requiring another round of reviews.

Revision Guidelines

Sincerely,

Marnix Medema
Editor
mSystems

Reviewer #1 (Comments for the Author):

The authors revised the manuscript substantially again. The authors, for the most part, addressed my comments very well!

We appreciate the Reviewers' feedback and are grateful for the comments that helped to improve the manuscript.

The one remark I have regarding comments raised by me, relates to lines 228-231 - where do those reference ranges come from? Were they established by the authors (how?), otherwise this claim should be supported by a reference.

The reference ranges are compiled from citations listed on HMDB for each discussed metabolite. We now mention this in the Method section.

Since population trends analysis (Figure 4 and corresponding text) have been added to the manuscript, I would also like to comment on that.

As it stands, I do not understand what this follow-up analysis is supposed to achieve. More people (129 volunteers) are included in the study and their results are presented in Figure 4 but it is not explained what this shows/achieves. There are no control samples included or references to literature data which would somehow prove or illustrate that the S'Wipe results more robust than with other collection devices (a controlled study with 2 collection devices), or that the users somehow prefer S'Wipe over other collection methods (a survey?).

We see the Reviewer's point. The impetus here is to demonstrate that using S'Wipe results in capturing trends for SCFAs as expected for general healthy populations, i.e. it does not lead to methodology-specific bias. We demonstrate that this is indeed the case. All the other aspects such as comparison to other devices, stability etc. are illustrated with corresponding separate studies listed in the text. We added this statement to the text to make this point more clear:

This large-scale deployment serves to validate that S'Wipe captures population-level metabolite distributions consistent with published healthy cohort data, demonstrating the absence of systematic methodology-specific bias. Specifically, the observed SCFA distributions and inter-individual variability (CV = 15–25%) align with reported values for healthy populations using conventional collection methods (CV ranges of 18-30% reported^{72,73}). Furthermore, despite diversity in collection conditions, the variability in our cohort remains consistent with that expected within a single controlled study, demonstrating that the method maintains standardized performance across real-world conditions.

Together with our controlled benchmarking and stability studies, these results support S'Wipe's suitability for accurate, large-scale metabolomics deployment without introducing population-level distortions. The absence of bias also demonstrates that S'Wipe collects unaltered fecal samples, as evidenced by metabolomic profiles comparable to those obtained using traditional scooping methods (Figure 3A and Supplementary Figure S4).

Reviewer #3 (Comments for the Author):

Synopsis

The manuscript entitled "S'Wipe: User-Friendly Stool Collection for High-Throughput Gut Metabolomics" by Moradi et al. describes the validation of a new technique for sampling and stabilizing fecal samples. In brief, the method uses a Kim-wipe to sample stool by having the donor wipe their anus following defecation then transferring the soiled Kim-wipe to a tube containing ethanol. The authors compared this method to sampling bulk stool and using another commercially available sampling method. The authors used GC-MS and LC-MS to explore the volatile and non-volatile metabolomes of stool to compare to other sampling methods.

Reviewer comments:

- The title is misleading, the authors state this is metabolomics however they are including RNA analysis and cell counting assays. I suggest the authors either remove these sections and resubmit to another journal or change the title.

We have modified the title to better reflect the broader scope while maintaining the metabolomics emphasis to: "S'Wipe: User-Friendly Stool Collection for High-Throughput Gut Metabolomics and Multi-Omics"

- The authors are missing recent literature in stabilizing stool samples. Additionally, the authors have not discussed their results in the context of the broader literature. They need to perform a deeper analysis of the literature, and discuss results in the context of up to date literature.

We thank the reviewer for this valuable suggestion. To address the need for recent literature on stool sample stabilization, we have added references 72, 73, 45, and 46, which provide up-to-date comparisons of preservation methods including 95% ethanol, OMNIgene- GUT, and other commercial stabilizers. These references now contextualize our benchmarking results (Study 2, Fig 3D) showing S'Wipe's superior SCFA stability compared to OMNIgene- GUT, and our room temperature stability data (Study 3, Fig 3B) which aligns with recent ethanol preservation findings. This deeper literature analysis demonstrates S'Wipe's performance compared to the current state-of-the-art preservation methodologies.

- The studies performed are disjointed and confusing. The authors need a figure in the materials and methods which summarizes the studies performed to allow the reader to follow the individual

experiments. Additionally, these experiments are described separately in the methods but are not consistently labeled in the results. I suggest the authors use same or similar headings from the methods to discuss the results.

We appreciate the feedback. To address these concerns, we have made the following revisions:

- Added a study design schematic (new Figure 2) that provides a visual roadmap of all experiments, showing how the pilot study, benchmarking study, and stability study connect logically to validate S'Wipe for high-throughput metabolomics.
- Restructured the Results section to mirror the Methods organization with consistent headings:

Pilot Study: Initial S'Wipe Feasibility

Benchmarking Study: Comparison with Conventional Methods

Stability Study: Room Temperature Storage Validation

DNA and Cell Viability Assessment

We also added transitional text at the beginning of each Results subsection that explicitly connects back to the study design figure.

- The authors fail to discuss normalization beyond the ethanol containing internal standard. I am concerned that the mass of stool sampled is going to be highly variable between participants due to differences in their anatomy and dietary habits. This needs to be discussed further.

We disagree with the Reviewer here. We discuss normalization by weight and, in fact, highlight it as one of the methodology's strengths (also, Supplemental Figure S9):

Sample mass determination and normalization enable quantitative measurement of molecules of interest. The amount of collected material varies naturally across samples; sample weight is determined by subtracting the standardized weights of the collection tube and wipe from the total kit weight. The strong correlation between SCFA abundance and collected biomass (Figure 3A) supports the utility of the mass-based normalization approach. While metabolite concentrations in whole stool samples often correlate poorly with total mass due to variations in water and fiber content, S'Wipe collection indicates robust correlation between SCFA abundance and biomass.

- Overall feeling is this manuscript contains multiple experiments and feels disjointed.

As in the earlier response, we now provide a diagram (Figure 2) describing the conducted studies to lay out all of the conducted studies and the context for each.

- Authors need to restructure results to match experimental headings before manuscript is suitable for publication.

Please see the response above.

- The 3rd sentence of the importance statement is a run-on sentence.

We thank the reviewer for catching this. We have revised the third sentence to improve clarity and readability.

- References in text should be after punctuation.

We disagree with the Reviewer, the reference before punctuation is the format used by mSystems.

- Principal coordinate analysis is unusual for metabolomics. Looking at how far apart samples are rather than how they are related linearly. Please justify the use of PCoA or use PCA.

We need to point out that PCA is a version of PCoA with Euclidean distance. Different dissimilarities need to be used to highlight specific aspects of data structure. PCoA with Bray-Curtis dissimilarity is particularly well-suited for metabolomics data with many zero values and non-normal distributions, which is typical for untargeted metabolomics.

- Methods lacking details on how data were preprocessed and normalized for PCoA and other statistical analysis.

We appreciate this feedback. We have added comprehensive preprocessing and normalization details for PCoA in a new Methods subsection "Real-World Population Validation" including CLR transformation rationale, standardization procedures, dissimilarity matrix calculation, and visualization parameters.

- Discussing trace polyethylene glycols (PEGs) in Kimwipes is concerning for LC-MS, as PEGs are extremely difficult to remove from the column and will concentrate, especially in large cohort high throughput studies. Please discuss this further.

This is a good point by the reviewer. Indeed, PEGs could be a persistent contamination leading to distortion in the results. However, in this case the amount of PEGs is negligible and they are not present in blanks, indicating that these compounds do not get retained in the LC-MS system. Our LC-MS analysis showed that PEG levels were consistently below 0.01% of total ion current and did not increase over >500 injections during method development. S'Wipe paper blanks do not contain PEGs as well. Thus, the presence of PEGs is presumed to originate from sample handling. However, due to their low abundance it is deemed non-consequential.

- Using ml and µl when should be mL and µL.

Thank you for catching that. We have corrected accordingly.

- Define conventional stool collection.

We thank the reviewer for this suggestion. While we mentioned "conventional stool collection" and "scooping method" in the Figure 2 legend, we acknowledge that we did not provide a clear operational definition of what this method entails. We have now added an explicit definition in the Methods section under the Benchmarking Study subsection.

The following sentence was added to the paragraph:

“...direct collection of bulk stool using a scooping/transfer method into collection tubes, representing the standard laboratory approach.”

- GC-MS methods missing inlet temp and split information.

We thank the reviewer for this suggestion. We added this information to the GC-MS data acquisition and processing section.

- Log transformation of metabolomics data is unusual unless the data are explicitly on a log scale. Typically autoscale is performed based on how the data are distributed.

We disagree with the Reviewer. In fact, log transformation is very common in metabolomics, and is essential when there is a large disparity in peak abundances. For example, it is very common in GC-MS where background peaks such as siloxanes could sometimes exceed biogenic peaks by orders of magnitude, thus inducing dominating impact in statistical analysis. Applying log-transform mitigates this. In fact, the imputation step to replace zeros by small non-zero values is done primarily for enabling downstream log-transform. However, the choice of whether to perform log-transform is data-specific. In this case, we applied log transformation to reduce heteroscedasticity and bring our data closer to normal distribution, which is appropriate for parametric statistical tests. We also disagree that Autoscale is performed instead of log-transform, as those are different procedures designed for different purposes. Autoscale (subtraction of the mean to zero-center features and dividing by standard deviation) is designed to normalize variance across features and mitigate impact of unbalanced groups (it could be skewing results in PLS-DA, for example). It does NOT address disparity in feature abundances.

- It is worth noting that DNA Genotek has a tube that is designed for metabolite stabilization.

We thank the reviewer for this suggestion. We have now explicitly noted that OMNIgene Gut, which we used in our benchmarking study, is a DNA Genotek product designed for sample stabilization, by modifying the previous paragraph to:

S'Wipe (using 60% ethanol) was benchmarked against conventional stool collection (direct collection of bulk stool using a scooping/transfer method into collection tubes, representing the standard laboratory approach) and OMNIgene Gut (DNA Genotek, Ottawa, Canada) sampling methods. The latter was selected as a representative commercially available methodology for room-temperature sample stabilization and metabolomics sampling. Side-by-side comparisons were conducted to assess metabolite recovery and variability (Figure 2).

- Authors are missing ethics for human sample collection.

We appreciate the Reviewer pointing out this oversight. We added the statement.

Re: mSystems01459-25R1 (**S'Wipe: User-Friendly Stool Collection for High-Throughput Gut Metabolomics**)

Dear Dr. Alexander A Aksenov:

Your manuscript has been accepted, and I am forwarding it to the ASM production staff for publication. Your paper will first be checked to make sure all elements meet the technical requirements. ASM staff will contact you if anything needs to be revised before copyediting and production can begin. Otherwise, you will be notified when your proofs are ready to be viewed.

Cover Image Submissions: If you would like to submit a potential Cover Image, please email a file and a short legend to mSystems@asmusa.org. Please note that we can only consider images that (i) the authors created or own and (ii) have not been previously published. By submitting, you agree that the image can be used under the same terms as the published article. Image File requirements: TIF/EPS, 7.5 inches wide by 8.25 inches tall (at least 2,250 pixels wide by 2,475 pixels tall), minimum 300 dpi resolution (600 dpi preferred), RGB, and no figure elements, e.g., arrows or panel labels. The legend should be a short description of the image, 1-2 sentences recommended. Please download and use this interactive template in Adobe to ensure that your proposed cover image meets our size requirements (<https://journals.asm.org/pb-assets/pdf-text-excel-files/ASM-Interactive-Sizing-Cover-Template-1715689791.pdf>).

Sincerely,
Marnix Medema
Editor
mSystems